# A Multistage Rigid-Affine-Deformable Network for Three-Dimensional Multimodal Medical Image Registration

**Anika Strittmatter** [1,2,*] , **Anna Caroli** [3] and **Frank G. Zöllner** [1,2]

1   Computer Assisted Clinical Medicine, Medical Faculty Mannheim, Heidelberg University, Theodor-Kutzer-Ufer 1-3, 68167 Mannheim, Germany; frank.zoellner@medma.uni-heidelberg.de
2   Mannheim Institute for Intelligent Systems in Medicine, Medical Faculty Mannheim, Heidelberg University, Theodor-Kutzer-Ufer 1-3, 68167 Mannheim, Germany
3   Bioengineering Department, Istituto di Ricerche Farmacologiche Mario Negri IRCCS, Via Gian Battista Camozzi 3, 24020 Ranica, BG, Italy; acaroli@marionegri.it
*   Correspondence: anika.strittmatter@medma.uni-heidelberg.de

**Abstract:** Multimodal image registration is an important component of medical image processing, allowing the integration of complementary information from various imaging modalities to improve clinical applications like diagnosis and treatment planning. We proposed a novel multistage neural network for three-dimensional multimodal medical image registration, which addresses the challenge of larger rigid deformations commonly present in medical images due to variations in patient positioning in different scanners and rigid anatomical structures. This multistage network combines rigid, affine and deformable transformations in three stages. The network was trained unsupervised with Mutual Information and Gradient L2 loss. We compared the results of our proposed multistage network with a rigid-affine-deformable registration with the classical registration method NiftyReg as a baseline and a multistage network, which combines affine and deformable transformation, as a benchmark. To evaluate the performance of the proposed multistage network, we used four three-dimensional multimodal in vivo datasets: three renal MR datasets consisting of T1-weighted and T2-weighted MR scans and one liver dataset containing CT and T1-weighted MR scans. Experimental results showed that combining rigid, affine and deformable transformations in a multistage network leads to registration results with a high structural similarity, overlap of the corresponding structures (Dice: $76.7 \pm 12.5$, $61.1 \pm 14.0$, $64.8 \pm 16.2$, $68.1 \pm 24.6$ for the four datasets) and a low level of image folding ($|J| \leq 0$: less than or equal to 1.1%), resulting in a medical plausible registration result.

**Keywords:** machine learning; image registration; multistage; deep learning; multimodal; medical images

## 1. Introduction

Multimodal image registration is a vital research field in medical image processing as it allows the integration of complementary information from various imaging modalities. This fusion can aid numerous clinical applications, such as improved diagnosis and treatment planning [1–3].

For instance, medical image registration is of great importance for the diagnosis and monitoring of renal diseases [4–6]. Among them, autosomal dominant polycystic kidney disease (ADPKD) is a chronic hereditary disorder that slowly leads to renal enlargement and eventually to end-stage renal disease (ESRD). Early detection and continuous monitoring of ADPKD progression are crucial for preventing or delaying ESRD. Medical image registration can be used to combine information from multiple imaging modalities, which can enhance the accuracy of ADPKD diagnosis and monitoring. Here, the total kidney volume (TKV) and the volumes and numbers of cysts are important biomarkers used to evaluate the progression of ADPKD [7]. Specifically, T1-weighted MR images are used to generate kidney segmentations to calculate the total kidney volume while T2-weighted MR images are used to create cyst segmentations to determine the cyst volume and the

number of cysts. By fusing the images of the two MR modalities and thereby combining complementary information, the determination of the two biomarkers can be significantly improved [8]. Furthermore, images from multiple time points can be fused, e.g., images before, during and after a therapy, to monitor the therapy's progress. For this, registration of the images is necessary.

Another example of a clinical application that benefits from image registration is the treatment of oligometastatic disease, which has poor outcomes with standard therapies due to the local lesion heterogeneity. Personalized image-guided therapies that consider tumor heterogeneity have shown potential in improving survival rates [9,10]. Typically, CT imaging is used for these interventions, while MRI is utilized to characterize tumor heterogeneity, which is often invisible in CT scans [11]. Differences in patient positioning in the two scanners, as well as the patient's respiration, result in the organs being deformed differently in the corresponding CT and MR images. Therefore, prior to fusion, the registration of CT and MR images is inevitable.

Traditionally, medical image registration is performed using conventional methods that involve iteratively updating the transformation parameters to optimize the similarity metrics. However, this approach is often time-consuming as each new image must be re-registered iteratively. To solve this problem, deep learning-based registration methods have been developed to speed up the registration process. In these methods, the network is trained once and subsequent registrations can be performed by passing the new images through the network, eliminating the need for time-consuming iterative methods [12].

In recent years, several neural networks for medical image registration have been published with a wide variety of architectures. These include, e.g., the use of multistage networks, in which different subnetworks are used [13–28]. In these approaches, an initial affine transformation is combined with a subsequent deformable transformation within a neural network in order to first reduce large global image displacements by the affine transformation subnetwork and then to correct smaller local image displacements by the deformable transformation subnetwork.

In medical image registration, image deformation also consists of larger rigid parts, for example, due to different patient positioning in different scanners or acquisitions. For instance, when planning and performing an image-guided biopsy as in [29], large rotations occur if the patient is supine during a preliminary examination but lateral during the image-guided biopsy due to needle placement. In addition, rigid structures such as bones or tumors that behave rigidly [30] are only or primarily deformed rigidly. Initially, a rigid transformation could be used to roughly reduce the large rigid displacements before correcting affine and deformable deformations. For instance, the NiftyReg algorithm reg_aladin [31] first applies rigid and then affine registration. In combination with the algorithm NiftyReg reg_f3d [32], a multistage registration with first rigid, then affine and subsequently deformable transformation is obtained.

However, to the best of our knowledge, no multistage network has yet been published that combines the three transformations rigid, affine and deformable. For this reason, we developed a multistage network for jointly rigid, affine and deformable transformation. This network can be trained end-to-end.

## 2. Background

The aim of image registration is to compute a transformation to spatially align two images, where the source (moving) image is registered to the reference (fixed) image. The computed transformation is then applied to the moving image to create the moved image. In medical image registration, three basic transformation types are applied: rigid, affine and deformable. A rigid transformation includes translation and rotation and is represented by

$$T_{rigid}(x) = Rx + t \tag{1}$$

where $R$ describes the rotation matrix, $x$ a point and $t$ the translation vector. For three-dimensional images, the rotation matrix and the translation vector require three parameters each. Thus, six parameters have to be calculated for a rigid transformation for three-dimensional images.

The affine transformation is composed of translation, rotation, scaling and shearing. Matrix multiplication is used to merge the single transformations, as described by [33]. For three-dimensional images, the affine transformation is represented by a $4 \times 4$ matrix that contains 12 parameters, including the translation parameters in the right column (parameters $a_{14}, a_{24}, a_{34}$), while the other nine parameters represent the combined values of rotation, scaling and shearing.

$$T_{affine} = \begin{bmatrix} a_{11} & a_{12} & a_{13} & a_{14} \\ a_{21} & a_{22} & a_{23} & a_{24} \\ a_{31} & a_{32} & a_{33} & a_{34} \\ 0 & 0 & 0 & 1 \end{bmatrix} \tag{2}$$

Nonlinear (deformable) transformations are used to model local deformations in images that rigid and affine transformations cannot capture. These transformations are complex and do not preserve straightness or parallelism. The transformation is represented by a deformation (displacement) field $\phi$:

$$T_{deformable}(x) = x + \phi(x) \tag{3}$$

Since deformable image registration is an ill-posed problem, regularization is necessary to ensure a smooth and plausible deformation field, e.g., by using penalty terms during the optimization of the transformation (iterative methods) or training of the registration network (deep learning-based) [34].

Conventional registration methods such as SimpleElastix [35] or NiftyReg [31,32] rely on solving a mathematical optimization model separately for each pair of images, which can be a slow and computationally intensive process. This involves iteratively maximizing a function that measures the similarity between the fixed and moved images. While some conventional methods can be accelerated using GPU implementations, this still requires a GPU for each registration process [36].

In contrast, neural networks offer a faster and more efficient approach to image registration. Instead of solving an optimization problem for each new pair of images, a neural network takes two n-dimensional images as an input and outputs the transformation in a single forward pass through the neural network [12]. The weights of the neural network are optimized through training, resulting in a globally optimized function that can be used for the registration of any test image pair without requiring an iterative optimization process. This approach saves time both on the CPU and GPU compared to conventional methods [12].

## 3. Methods

### 3.1. Network Architectures

We developed a multistage rigid-affine-deformable neural network that comprises three subnetworks, namely a rigid, affine and deformable registration network.

We used a modified variant of the affine network proposed by [15], since it achieved the best registration results in our datasets as demonstrated previously [37]. The network architecture was modified to allow for a rigid transformation in addition to the original affine transformation. The rigid and affine subnetwork is composed of five convolutional blocks (Figure 1). Each contains a three-dimensional convolutional layer with a kernel size of $3 \times 3 \times 3$, a stride of 1 and padding same, a maxpooling layer with a factor of 2 and a LeakyReLU activation layer. A dense layer is then used to calculate the 6 or 12 parameters required for a three-dimensional rigid or affine transformation.

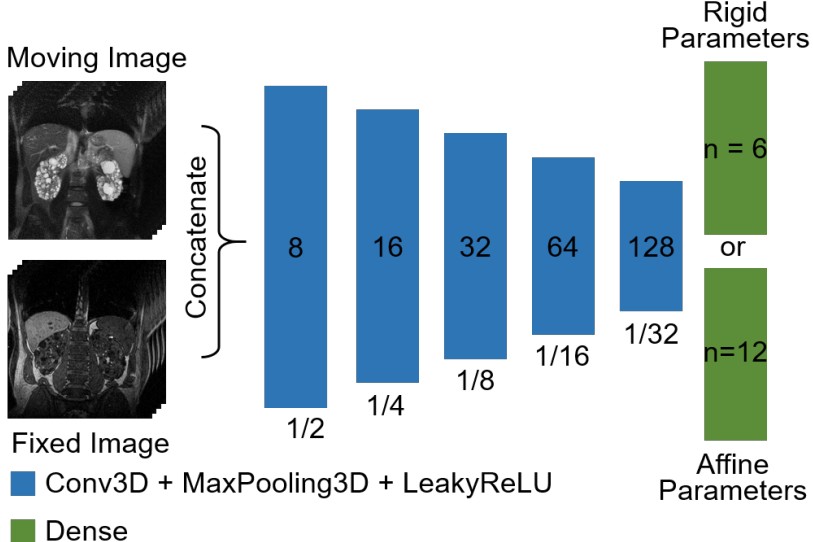

**Figure 1.** Architecture of the rigid and affine subnetworks. Rigid subnetwork: 6 neurons compute 6 parameters for rigid transformation. Affine subnetwork: dense layer computes 12 parameters for affine transformation.

The third subnetwork network (Figure 2a) performs a deformable transformation. Here, we used the VoxelMorph-2 network [12,38].

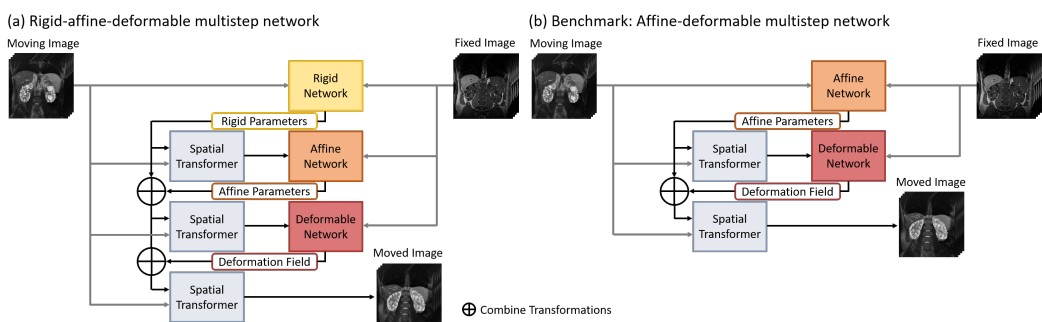

**Figure 2.** (**a**) Architecture of the multistage network with rigid, affine and deformable subnetworks. The individual subnetworks are connected in series. The rigid, affine and deformable transformation is combined and then applied to the moving image by a spatial transformer [39]. This result image is the final moved image. (**b**) Architecture of the benchmark with affine and deformable subnetworks.

To form the aforementioned rigid-affine-deformable multistage network (see Figure 2a), the individual subnetworks are connected in series. The rigid subnetwork calculates a rigid transformation, which is then applied to the moving image with a spatial transformer [39]. The result image is used as an input for the affine transformation subnetwork. The calculated affine transformation is then combined with the rigid transformation. The result transformation is then applied to the original moving image and the result image then serves as an input for the deformable subnetwork. The deformation field calculated by the deformable subnetwork is then combined with the rigid and affine transformation. The result transformation is applied to the moving image using the spatial transformer. This result image is the final moved image in our proposed multistage rigid-affine-deformable network (Figure 2a).

To facilitate the combination of the different single transformations, we calculated the deformation field (vectors) from the rigid and affine transformations. Two transformations can then be combined by adding their deformation fields.

### 3.2. Benchmark Neural Network

As a benchmark, we used a neural network with two subnetworks (Figure 2b). The first subnetwork performs an affine transformation, while the second subnetwork a deformable transformation. For the affine and deformable subnetworks, we used the same architectures as for the affine and deformable subnetworks from our multistage rigid-affine-deformable network. The difference between our proposed multistage rigid-affine-deformable network and the benchmark network therefore lies in a preceding rigid subnetwork. Thus, in our experiments, we examined the influence of a preceding rigid subnetwork on the registration result.

### 3.3. Training Setting for Neural Networks

We trained each subnetwork separately while keeping the weights of the preceeding stages fixed. For our proposed multistage network, we first trained the rigid subnetwork and then added the affine subnetwork. We froze the layers of the rigid subnetwork so that only the weights of the affine subnetworks were updated during the training of the multistage rigid-affine network. After training, we added the deformable subnetwork and trained this subnetwork while preserving the weight of the preceeding stages. We then unfroze the layers of all subnetworks and finetuned all subnetworks in the multistage network jointly. Similarly, for the benchmark CNN, we first trained the affine subnetwork, then froze its layers, added the deformable stage and then trained the network. We then unfroze the layers of the affine subnetwork and finetuned the two subnetworks jointly. In the experiments, we trained the networks unsupervised with the Mutual Information loss of the fixed image ($F$) and the predicted moved image ($M$)

$$MI(F, M) = I(F; M) = H(F) + H(M) - H(F, M) \tag{4}$$

with the entropy $H$. Additionally, we used the Gradient L2 loss as a regularization of the deformation field $\phi$:

$$L_2(\phi) = \sum_{p \epsilon \Omega} ||\nabla \phi(p)||^2 \tag{5}$$

The image similarity loss (Mutual Information, $L_{MI}$) and the regularization loss of the deformation field (Gradient L2, $L_2$) are then combined by the lambda parameter $\lambda$:

$$L(F, M) = L_{MI}(F, M) + \lambda L_2(\phi) \tag{6}$$

We used the Keras Hyperband Tuner [40] for optimizing the learning rate, optimisation function and $\lambda$-parameter for each of the proposed models. The parameter configuration that resulted in the highest Mutual Information was selected; the networks were trained with this configuration. For all experiments with the datasets, intensity normalization to the range [0, 1] was used. We conducted five-fold cross-validation and trained the networks for 200 epochs using a batch size of one. We applied early stopping of the training if the validation loss did not improve after five epochs. We saved the weights of the network that achieved the highest accuracy (highest Mutual Information) on the validation data. The fixed and moving images were resampled to $256 \times 256 \times 64$ voxels and a voxel spacing of $2 \times 2 \times 4$ mm$^3$ using trilinear interpolation. The result transformation was resized and applied to the full-resolution images. The experiments were performed in Python 3.7 using Tensorflow [41] on a workstation equipped with an NVIDIA RTX A5000 with 24 GB GPU memory and an NVIDIA RTX A6000 with 48 GB GPU memory (NVIDIA, Santa Clara, CA, USA).

### 3.4. Baseline Method

As a baseline, we applied the classical registration method NiftyReg. First, the moving image was registered rigidly and then affinely with the algorithm reg_aladin [31]. The result image was then transformed deformably with the algorithm reg_f3d [32]. In addition to

the fixed and moving image, both algorithms used the segmentation of the fixed and moving image to determine the structure of interest [31]. Additionally, we transformed the segmentation of the moving image with the transformation calculated by NiftyReg. The result is the segmentation of the moved image.

*3.5. Datasets*

We assessed the performance of the proposed multistage network using four multimodal three-dimensional in vivo datasets. The imaging parameters of the datasets are listed in Table 1.

**Table 1.** Dataset statistics.

| Dataset | Modality | Volumes | Volume Size | Resolution (mm$^3$) |
|---|---|---|---|---|
| First NIDDK dataset | T1-weighted MR scans | 100 | $256 \times 256 \times [30, 80]$ | $1.41 \times 1.41 \times 3.06$ |
| | T2-weighted MR scans | 100 | $256 \times 256 \times [12, 30]$ | $1.39 \times 1.39 \times 9.02$ |
| Second NIDDK dataset | T1-weighted MR scans | 100 | $256 \times 256 \times [30, 80]$ | $1.41 \times 1.41 \times 3.06$ |
| | T2-weighted MR scans | 250 | $256 \times 256 \times [10, 51]$ | $1.38 \times 1.38 \times 3.00$ |
| External kidney dataset | T1-weighted MR scans | 41 | $512 \times 512 \times [40, 66]$ | $0.74 \times 0.74 \times 4.00$ |
| | T2-weighted MR scans | 41 | $256 \times 256 \times [30, 70]$ | $1.46 \times 1.46 \times 4.02$ |
| M$^2$OLIE dataset | CT scans | 47 | $512 \times 512 \times [25, 132]$ | $0.78 \times 0.78 \times 2.04$ |
| | T1-weighted MR scans | 73 | $[190, 440] \times [288, 640] \times [52, 120]$ | $1.27 \times 1.27 \times 3.77$ |

3.5.1. NIDDK

As the first dataset, we used in vivo images from the Consortium for Radiologic Imaging Studies of Polycystic Kidney Disease (CRISP) study, provided by the National Institute of Diabetes and Digestive and Kidney Disease (NIDDK), National Institutes of Health, USA [42]. We utilized 100 datasets (i.e., a total of 100 T1-weighted and 100 T2-weighted scans) of patients with different stages of ADPKD (for details see [43]) obtained from the NIDDK database.

Manual segmentation of the kidneys was performed by two physicians on the T1- and T2-weighted coronal MR scans using an in-house developed annotation tool based on MeVisLab SDK (MeVis Medical Solutions, Inc., Bremen, Germany). Furthermore, the annotation tool was used to evaluate the inter-user agreement of the kidney segmentations. This resulted in a mean inter-user agreement of $0.91 \pm 0.06$ (Dice) for the kidney segmentations of the 100 T1-weighted MR scans and a mean inter-user agreement of $0.92 \pm 0.02$ (Dice) for the kidney segmentations of the 100 T2-weighted MR scans.

The second dataset also consists of T1- and T2-weighted MR scans taken from the NIDDK database. The second dataset contains the 100 T1-weighted MR scans of our first dataset. As a second modality, we used 250 T2-weighted MR scans with a slice thickness of 3 mm.

3.5.2. External Kidney Dataset

The third dataset contains 41 in vivo T1-weighted and T2-weighted renal MR scans from patients with ADPKD, acquired from the Azienda Socio-Sanitaria Territoriale (ASST) Papa Giovanni XXIII, Bergamo, Italy, in the context of the observational EuroCYST study (ClinicalTrials.gov Identifier: NCT02187432). The T1- and T2-weighted MR scans were acquired in coronal orientation. For the T1-weighted and T2-weighted MR scans, segmentation was performed manually. For the T1-weighted MR scans, kidney segmentations were obtained to quantify the total kidney volume. For the T2-weighted scans, kidney cyst segmentations were created.

In our experiments, in the first, second and third dataset, the T2-weighted MR scans were registered to the T1-weighted MR scans.

### 3.5.3. M²OLIE

The fourth dataset comprises CT and T1-weighted MR liver volumes of 39 patients. These were acquired as part of the "Mannheim Molecular Intervention Environment" (M²OLIE) project. The dataset includes 47 CT scans and 73 T1-weighted MR scans with in-phase and opposed-phase in transversal orientation. Liver segmentations were created using two separate segmentation networks for the CT and MR scans (three-dimensional U-Nets [44]). The networks were trained on the 3D-IRCADb-01 [45] and Combined Healthy Abdominal Organ Segmentation (CHAOS) [46] datasets, respectively. We then finetuned both networks using 20 segmentations of the CT scans and 20 segmentations of the MR scans of the M²OLIE dataset, which we created manually in the Medical Imaging Interaction Toolkit (MITK) [47]. The segmentations generated by the networks were visually inspected to ensure that they were all acceptable. In our experiments, the MR scans were registered to the CT scans.

### *3.6. Evaluation Metrics*

The performance of the proposed multistage network was evaluated by conducting experiments on the multimodal datasets. The evaluation was based on volumes (three-dimensional images and segmentations). The degree of overlap between the anatomical segmentations of the moved image ($S_M$) and the fixed image ($S_F$) was determined using the Dice coefficient.

Additionally, we evaluated the level of image folding due to the deformable transformation by calculating the Jacobian determinant ($\det(J)$ or $|J|$) of each point of the deformation field. In cases where the Jacobian determinant is equal to 1, the volume remains unaltered, if it exceeds 1, there is volumetric expansion, and if it falls between 0 and 1, there is volumetric shrinkage. A Jacobian determinant of $\leq 0$ means that folding has occurred here. Image folding is anatomically impossible in medical images [22]. Therefore, the aim is a registration with a deformation field where the Jacobian determinant for each point is $>0$. For rigid and affine transformations, no folding can occur. The number of Jacobian determinants $\leq 0$ is therefore 0.

We applied a paired *t*-test (for normal distribution) or a Wilcoxon signed-rank test (for non-normal distribution) to determine the significance of the improvement in the Dice coefficient and number of Jacobian determinants $\leq 0$ after registration compared to the data before applying the registration (sub)network. The null hypothesis, which states that registration by the neural networks does not improve the two metrics, is rejected if $p < 0.05$.

For visual evaluation of the spatial alignment before and after registration, the fixed, moving and moved image were examined. Additionally, the moving/moved and fixed images were color overlaid to form a "Composite Image". If available, the segmentations of the moving/moved and fixed images were also color-overlaid to form a "Composite Segmentation". In addition, the deformation field was displayed in RGB colors. The three color channels represent the deformation in the three spatial directions of three-dimensional images and transformations. The color red corresponds to a displacement on the y-axis (height), green on the x-axis (width) and blue on the z-axis (slice direction). This enables the identification of areas with strong deformation and a check on whether the deformation is smooth within the individual organs and structures. For areas that are displayed in gray values, the same deformation is applied in all three spatial directions.

### *3.7. Experiments*

#### 3.7.1. Experiment I

First, we trained our proposed multistage rigid-affine-deformable network with the first NIDDK dataset and compared its registration results with those of the benchmark network. This comparison aimed to evaluate the effect of using a preceding rigid subnetwork on the registration result.

We trained our proposed multistage network (Figure 2a) as described in Section 3.3 and Table 2. First, all subnetworks were trained separately, i.e., the weights of all previous stages were frozen. We then finetuned the whole multistage network jointly. We trained the benchmark network similarly.

**Table 2.** Hyperparameter setting for the training of the different subnetworks and finetuning of the whole benchmark network and proposed multistage-network (row 3 and 7), with the first NIDDK dataset used.

| Network | | Trained Subnetwork | Learning Rate | Optimizer Function | $\lambda$-Parameter |
|---|---|---|---|---|---|
| Benchmark | Affine | Affine | $3 \times 10^{-4}$ | Adam | 0.8 |
| | Affine-Deformable | Deformable | $2 \times 10^{-4}$ | Adam | 0.7 |
| | Affine-Deformable Finetuned | All | $3 \times 10^{-4}$ | Adam | 1 |
| Proposed Multistage Network | Rigid | Rigid | $3 \times 10^{-4}$ | Adam | 0.6 |
| | Rigid-Affine | Affine | $5 \times 10^{-4}$ | Adam | 0.7 |
| | Rigid-Affine-Deformable | Deformable | $5 \times 10^{-4}$ | Adam | 1 |
| | Rigid-Affine-Deformable Finetuned | All | $3 \times 10^{-4}$ | Adam | 0.8 |

We then applied the trained networks on the second NIDDK dataset (T1-weighted and T2-weighted MR images with a slice thickness of 3 mm).

### 3.7.2. Experiment II

In addition, we applied the networks trained on the first NIDDK dataset to the external kidney dataset to determine whether the proposed multistage network also produces good registration results on a before-unseen dataset.

To further evaluate the performance of the proposed multistage network, we augmented the external kidney dataset, i.e., applied artificial transformations and investigated if these transformations could be corrected with the proposed multistage network. We created an augmented dataset comprising 410 T1-weighted and T2-weighted MR volumes. This dataset contains the original scans and, additionally to each T1-weighted or T2-weighted MR scan, 9 augmented volumes. These volumes were augmented with the transformations described in Table 3. Translation, rotation, scaling and deformation were applied with a probability of 40% each. For the deformation, the Volumentations toolbox [48] was applied with the deformation limits (0, 0.25). The segmentations were transformed accordingly.

**Table 3.** Parameters for data augmentation performed on the T1-weighted and T2-weighted MR scans of the external kidney dataset.

| Parameter | Value |
|---|---|
| Translation (fraction of image size) | $(-10, 10)$ |
| Rotation (degrees) | $(-10, 10)$ |
| Scaling (factor) | $(0.9, 1.1)$ |
| Deformation (parameter deformation_limits) | $(0, 0.25)$ |

We finetuned the networks (benchmark and proposed multistage network) trained on the first NIDDK dataset on this augmented external kidney dataset. First, we froze the weight of the deformable subnetwork of the benchmark and the multistage network. So, we finetuned the rigid or rigid and affine subnetworks (benchmark: learning rate of $1 \times 10^{-4}$, Adam optimizer and a $\lambda$-parameter of 0.9; multistage network: learning rate of $2 \times 10^{-4}$, Adam optimizer and a $\lambda$-parameter of 1). Then, we unfroze the weights of the deformable subnetwork and jointly finetuned all subnetworks (benchmark: learning rate of $1 \times 10^{-4}$, Adam optimizer and a $\lambda$-parameter of 10; multistage network: learning rate of $2 \times 10^{-4}$, Adam optimizer and a $\lambda$-parameter of 10).

### 3.7.3. Experiment III

We finetuned our proposed multistage network (trained on the first NIDDK dataset) with the M$^2$OLIE liver dataset (CT and MR scans) to demonstrate that the multistage network is generalisable and can be used for further applications, such as the registration of CT and MR scans of the liver.

We finetuned all subnetworks jointly without freezing weights of individual subnetworks. We finetuned the benchmark with a learning rate of $2 \times 10^{-4}$, Adam optimizer and a lambda parameter of 4.5. The multistage network was finetuned with a learning rate of $3 \times 10^{-4}$, Adam optimizer and a lambda parameter of 4.5.

## 4. Results

We demonstrate the effectiveness of our proposed multistage model through extensive experiments using the multimodal three-dimensional datasets described in Section 3.5. The experiments are described in Section 3.7.

### 4.1. Experiment I

Table 4 shows the registration results. The first stage of the multistage network (rigid subnetwork) and the first stage of the benchmark (affine subnetwork) yielded similar Dice coefficients. The rigid-affine subnetwork of the multistage network produced registration results with a significantly ($p < 0.05$) higher Dice coefficient than the affine subnetwork of the benchmark and a similar Dice coefficient as the baseline NiftyReg. The multistage rigid-affine-deformable network (separately trained subnetworks, row eight in Table 4) and the finetuned variant (row nine in Table 4) yielded similar Dice coefficients as the affine-deformable benchmark networks (separately trained and jointly finetuned, row four and five in Table 4). All stages of the benchmark and the proposed multistage network (Table 4 row three to row nine) yielded registration results with a significantly higher Dice coefficients than before registration (Table 4 row one). The proposed multistage networks computed transformations with a significantly ($p < 0.05$) lower level of image folding than the benchmark network ($|J| \leq 0$ in Table 2, row eight and nine vs. row four and five). By jointly finetuning the multistage network, the number of $|J| \leq 0$ was significantly reduced (row eight vs. row nine in Table 4). The results of the significance tests for the benchmark and proposed multistage network are listed in Table 5.

**Table 4.** The Dice coefficient and Jacobian determinant ($|J|) \leq 0$ results (mean ± SD) of the first NIDDK dataset of the baseline and each stage of the benchmark and our proposed multistage network. Dice coefficient: higher value is better (maximum is 100%); $|J| \leq 0$: lower value is better (minimum is 0%).

| Network | | DICE (%) | $|J| \leq 0$ (%) |
|---|---|---|---|
| Before Registration | | 67.6 ± 18.1 | - |
| Baseline NiftyReg | | 70.9 ± 24.5 | 0.2 ± 0.6 |
| Benchmark | Affine | 69.1 ± 16.5 | 0 ± 0 |
| | Affine-Deformable | 76.4 ± 12.4 | 0.7 ± 0.8 |
| | Affine-Deformable Finetuned | 76.4 ± 12.7 | 0.7 ± 0.8 |
| Proposed Multistage Network | Rigid | 68.7 ± 17.3 | 0 ± 0 |
| | Rigid-Affine | 70.9 ± 14.2 | 0 ± 0 |
| | Rigid-Affine-Deformable | 76.7 ± 11.4 | 0.6 ± 0.7 |
| | Rigid-Affine-Deformable Finetuned | 76.7 ± 12.5 | 0.5 ± 0.7 |

Examples of registration results can be seen in Figure 3. Before registration, the abdomen in the T1-weighted (fixed) and T2-weighted (moving) MR images is not well aligned. Registration with the baseline NiftyReg (second row in Figure 3) improved the structural similarity and the overlap of kidney segmentations. The rigid subnetwork (row six in Figure 3) significantly improved the spatial alignment compared to before registration. When adding the affine subnetwork, the overlap of the kidney segmentations is further

improved compared to the rigid subnetwork (row six and seven in Figure 3). However, other structures, such as the liver, do not overlap well in the fixed and moved image, as is visible in the composite image. By adding the deformable subnetwork (row eight in Figure 3), the structural similarity of all structures is improved, especially visible in the liver in the composite image. Jointly finetuning the multistage network further increased the overlap of all structures (clearly visible in the liver). Comparing the registration result of the jointly finetuned multistage network (row nine) with the jointly finetuned benchmark (row five) demonstrates improved spatial alignment of all corresponding structures, especially visible in the liver, using the proposed multistage network. The deformations within the organs are smooth (sixth column in Figure 3). The largest deformations are located at the organ boundaries, especially at the liver and spleen.

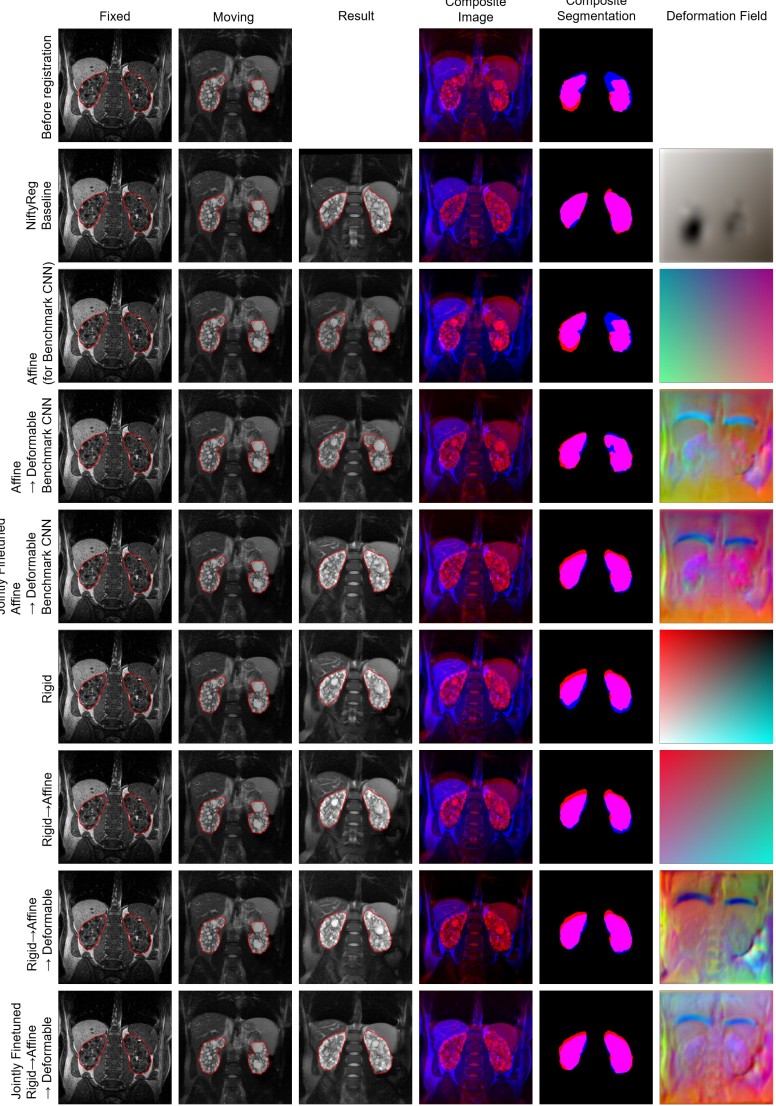

**Figure 3.** Example results of the registration of two volumes from the first NIDDK dataset. The images show central slices of the coronal plane. The fixed images, moving images and result (moved) images are overlaid by the kidney segmentations (column 1–3). The resulting composites for image and segmentation show the fixed data in blue and the moving/moved data in red. The deformation field is visualized in RGB channels.

We applied the trained networks on the second NIDDK dataset. The proposed multistage network yielded transformations with significantly ($p < 0.05$) less image folding than the benchmark ($|J| \leq 0$ (mean ± std): multistage network: 4.8 ± 2.1, benchmark: 6.1 ± 2.1).

Examples of registration results of the second NIDDK dataset can be seen in Figure 4. Before the registration, the liver especially is not well aligned in the fixed and moving image (white arrow in Figure 4). Registration with the baseline did not improve the spatial alignment the corresponding structures. The overlap of the liver in the fixed and moved image did not increase. The benchmark and multistage CNN produced registration results with good alignment. Within the organs, the deformations were smooth.

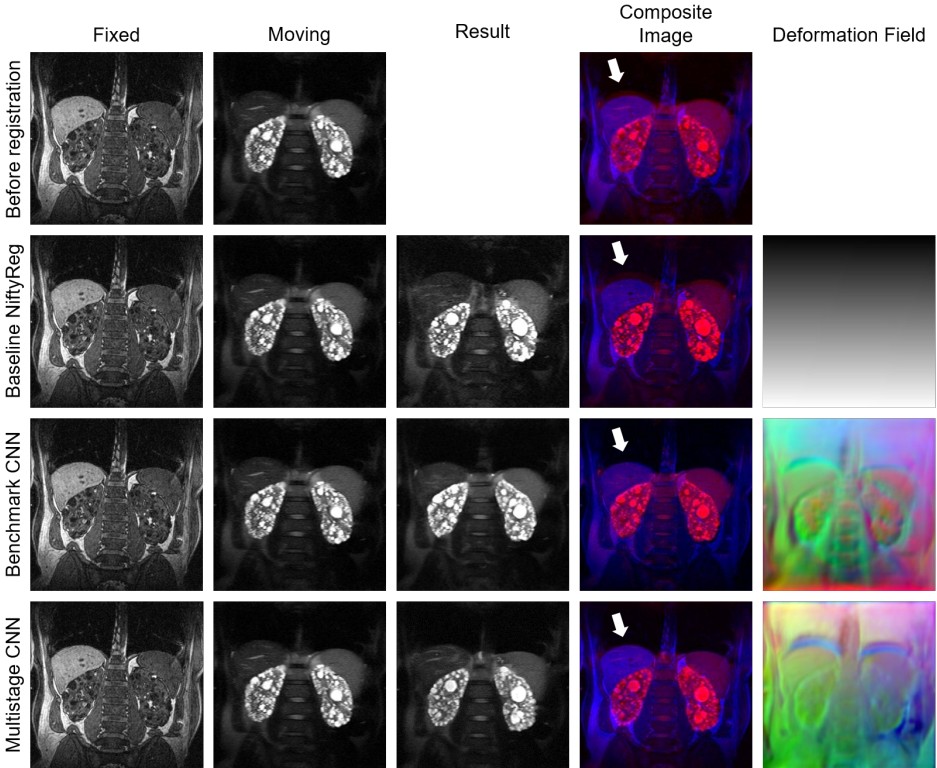

**Figure 4.** Example results of the registration of two volumes from the second NIDDK dataset. The images show central slices of the coronal plane. The resulting composites for image and segmentation show the fixed data in blue and the moving/moved data in red. The arrow in the composite image indicates the area with misalignment between the fixed and moving image. The deformation field is visualized in RGB channels.

**Table 5.** *p*-values of the significance tests (paired *t*-test or a Wilcoxon signed-rank test, *p* < 0.05) for the Dice coefficient and Jacobian determinant ($|J|$) ≤ 0 of the benchmark and proposed multistage network. Metrics with a significant difference (*p* < 0.05) are indicated in bold.

| Proposed Multistage Network | Benchmark | | | | | |
| --- | --- | --- | --- | --- | --- | --- |
| | Affine | | Affine-Deformable | | Affine-Deformable Finetuned | |
| | **DICE** | $\mathbf{\|J\| \leq 0}$ | **Dice** | $\mathbf{\|J\| \leq 0}$ | **Dice** | $\mathbf{\|J\| \leq 0}$ |
| Rigid | 0.0784 | - | | | | |
| Rigid-Affine | **0.0048** | - | | | | |
| Rigid-Affine-Deformable | | | 0.3125 | **<0.00001** | | |
| Rigid-Affine-Deformable Finetuned | | | | | 0.43675 | **<0.00001** |

### 4.2. Experiment II

Table 6 shows the registration results. Registration with the baseline NiftyReg improved the Dice coefficient. The benchmark and the proposed multistage network yielded similar Dice coefficients as before registration. Additionally, the benchmark and the proposed multistage network produced a similar level of image folding ($|J| \leq 0$).

**Table 6.** The Dice coefficient and Jacobian determinant (|J|) $\leq 0$ results (mean ± SD) of the external kidney dataset without and with augmentation of the baseline, benchmark and proposed multistage network. Dice coefficient: higher value is better (maximum is 100%), |J| $\leq 0$: lower value is better (minimum is 0%).

| Network | External Kidney Dataset without Augmentation | | External Kidney Dataset with Augmentation | |
|---|---|---|---|---|
| | DICE (%) | \|J\| <0 (%) | DICE (%) | \|J\| <0 (%) |
| Before Registration | 63.9 ± 11.3 | - | 45.1 ± 16.9 | - |
| Baseline NiftyReg | 67.3 ± 13.9 | 0.1 ± 0.2 | 59.5 ± 22.5 | 0.2 ± 1.0 |
| Benchmark | 62.4 ± 13.7 | 0.9 ± 0.4 | 62.5 ± 15.6 | 0.0 ± 0.0 |
| Proposed Multistage Network | 61.1 ± 14.0 | 1.1 ± 0.6 | 64.8 ± 16.2 | 0.0 ± 0.0 |

Examples of registration results can be seen in Figure 5. The fixed and moving image already show a reasonably good spatial alignment before registration; only fine structures in the kidney need to be corrected. The registration with the Baseline NiftyReg further improved the overlap of the kidneys. The benchmark and the proposed multistage network yielded similar results as before the registration. The deformation field of the multistage network shows smooth deformations.

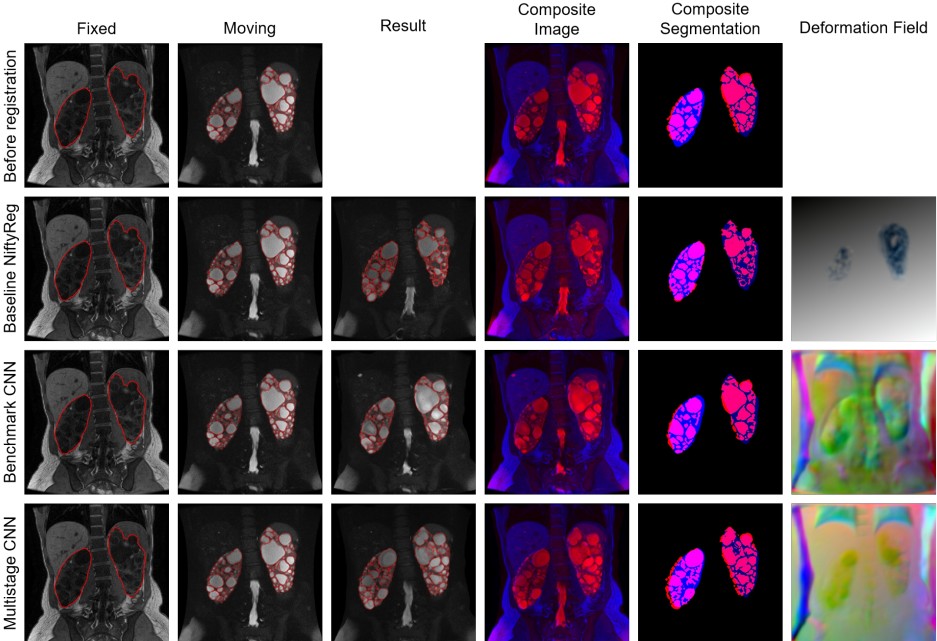

**Figure 5.** Example results of the registration of two volumes from the external kidney dataset. The images show central slices of the coronal plane. The fixed images are overlaid by the kidney segmentations and the moving images and result (moved) images are overlaid by the kidney cyst segmentations (column 2 and 3). The resulting composites for image and segmentation show the fixed data in blue and the moving/moved data in red. The deformation field is visualized in RGB channels.

Table 6 shows the registration results of the neural networks, which were finetuned and evaluated on the augmented external kidney dataset. The benchmark and the proposed multistage network yielded a significantly ($p < 0.05$) higher Dice coefficient than the baseline and registration with the proposed multistage network, which resulted in a significantly higher Dice coefficient than the benchmark. In addition, for the benchmark and the proposed multistage network, the level of image folding (|J| < 0) is significantly lower than for the baseline. For the benchmark and the proposed multistage network, the |J| < 0 is smaller than 0.05.

Examples of the registration results can be seen in Figure 6. Before registration, the corresponding images are not well aligned, especially visible at the spleen in the column "Compos-

ite Image" and at the segmentations in the column "Composite Segmentations". The baseline, benchmark and multistage network produced registration results with good spatial alignment of all corresponding structures. And the networks produced smooth deformations.

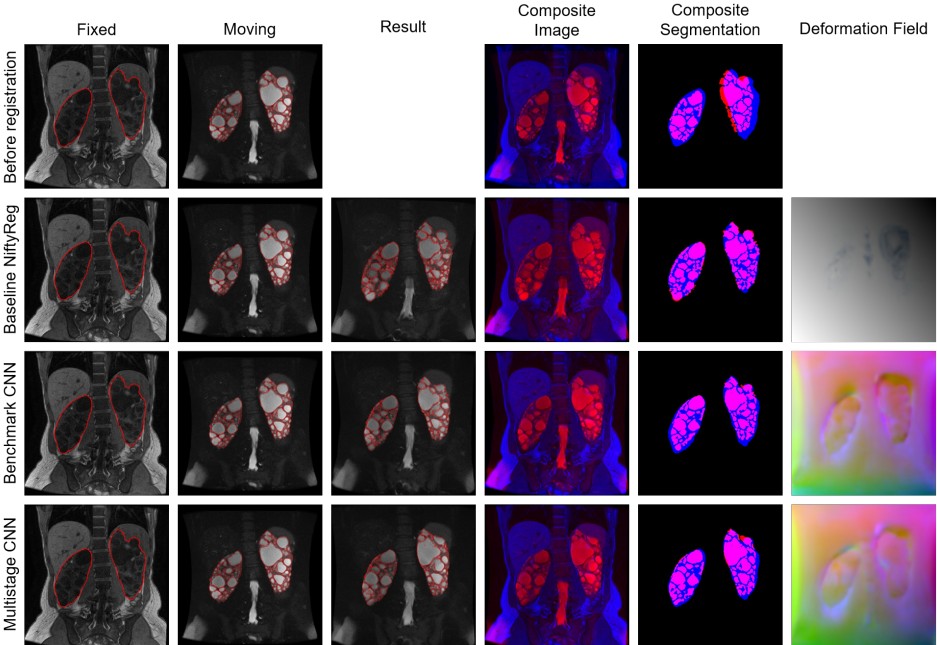

**Figure 6.** Example results of the registration of two volumes from the external augmented kidney dataset. The images show central slices of the coronal plane. The fixed images are overlaid by the kidney segmentations and the moving images and result (moved) images are overlaid by the kidney cyst segmentations (column 2 and 3). The resulting composites for image and segmentation show the fixed data in blue and the moving/moved data in red. The deformation field is visualized in RGB channels.

### 4.3. Experiment III

The results of the registration are shown in Table 7. The benchmark and our proposed multistage network yielded significantly ($p < 0.05$) higher Dice coefficients than the baseline. And registration with the proposed multistage network resulted in a significantly higher Dice coefficient than the benchmark. The level of image folding ($|J| < 0$) is similar for the benchmark and the proposed multistage network.

**Table 7.** The Dice coefficient and Jacobian determinant ($|J|$) $\leq 0$ results (mean ± SD) of the M$^2$OLIE liver dataset of the baseline, the benchmark and our proposed multistage network. Dice coefficient: higher value is better (maximum is 100%), $|J| \leq 0$: lower value is better (minimum is 0%).

| Network | DICE (%) | $|J| \leq 0$ (%) |
|---|---|---|
| Before Registration | 54.4 ± 19.9 | - |
| Baseline NiftyReg | 63.3 ± 25.5 | 0.0 ± 0.1 |
| Benchmark | 66.6 ± 24.0 | 0.1 ± 0.1 |
| Proposed Multistage Network | 68.1 ± 24.6 | 0.1 ± 0.1 |

Example registration results are presented in Figure 7. A large rigid part of the deformation is visible in the images before registration, as the fixed image is shifted downwards compared to the moving image. Registration with the baseline improved the overlap of the liver segmentations; however, the moved image is rotated incorrectly. Here, the rigid or affine algorithm of the baseline computed an incorrect transformation. Registration with the benchmark improved the spatial alignment and overlap of the liver segmentations. However, in the moved image, the torso on the right side and parts of the

right arm were deformed in a medically implausible way. This area is marked by a white arrow. The registration with the proposed multistage network also produced a medically implausible deformation on the right side of the torso (white arrow in result image in the fourth row). Apart from this area, the overlap of the corresponding structures improved following the registration. The area of the medical implausible deformation is also visible in the deformation field (white arrow in column six in Figure 7).

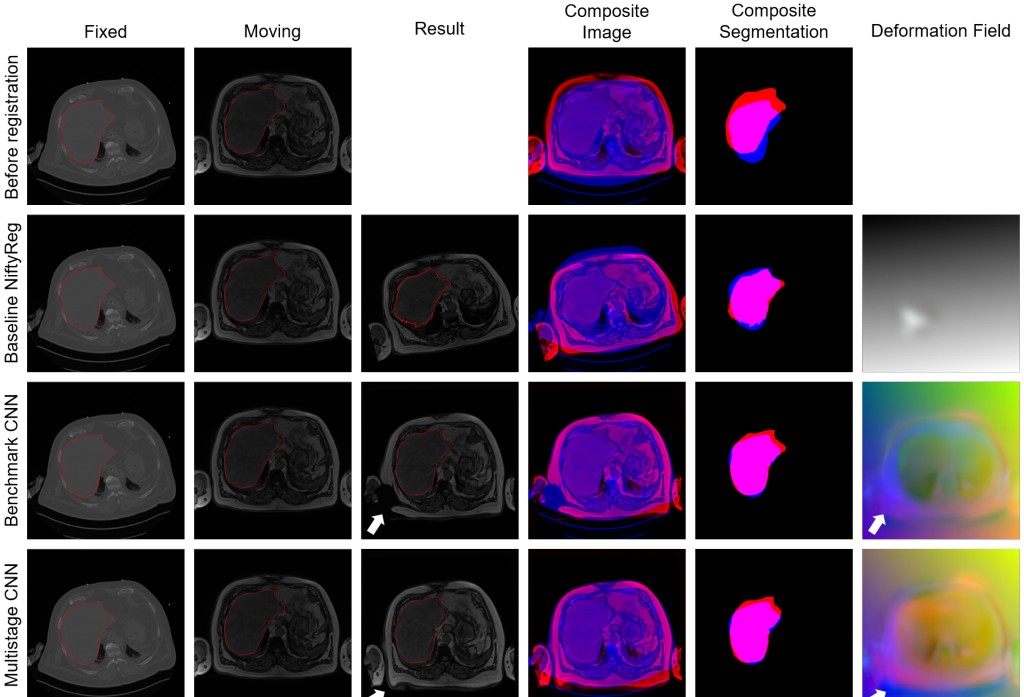

**Figure 7.** Example results of the registration of two volumes from the M$^2$OLIE liver dataset. The images show central slices of the coronal plane. The fixed images, moving images and result (moved) images are overlaid by the liver segmentations (column 1–3). The resulting composites for image and segmentation show the fixed data in blue and the moving/moved data in red. The arrow in the result image indicates the area with medical implausible deformation. The deformation field is visualized in RGB channels.

## 5. Discussion

We proposed a multistage end-to-end image registration neural network comprising rigid, affine and deformable transformation. By using an affine-deformable network as a benchmark in our experiments, we investigated the influence of a preceding rigid subnetwork on the registration result.

### 5.1. Comparison to Published Multistage Registration Methods

In classical (iterative) methods, approaches to multistage registration (affine and deformable or rigid-affine-deformable) exist. For instance, the SimpleElastix [35] documentation describes a non-rigid registration. Here, the affine algorithm is applied first and the bspline (deformable) algorithm afterwards. In the iterative registration method NiftyReg [31,32], the algorithm reg_aladin applies a rigid and affine registration by default. Thus, by connecting reg_aladin (rigid and affine) and the deformable registration with reg_f3d, a multistage rigid-affine-deformable registration is obtained. The ANTS (Advanced Neuroimaging Tools) [49] toolbox provides an elastic registration that uses affine and deformable transformations. In addition, a variant of the symmetric image normalization method (SyN) combines rigid, affine and deformable transformations (SyNRA).

However, these classical methods are often time-consuming as each new image pair requires iterative registration. Thus, the registration of a single image pair takes several

minutes, depending on the registration method and the hardware used (CPU or GPU). With neural networks, the registration process speeds up. In comparison, our proposed multistage network takes under a second to register an image pair on CPU or GPU.

Several neural networks have been published with multistage approaches for medical image registration [13–28]. Refs. [13–21,23,28] apply neural networks with two stages. In the first stage/subnetwork, an affine transformation is calculated, whereas in the second stage/subnetwork, a deformable transformation is calculated. Refs. [22,24–26] also utilize an affine network as the first subnetwork, but this is followed by a cascade of multiple deformable subnetworks. And Ref. [27] applies Generative Adversarial Networks (GANs) for affine-deformable transformation. The generator consists of a modality translator (for multimodal image registration), an affine-transformation regressor and a nonlinear-deformation regressor.

However, only affine and deformable transformations were applied in these papers. To the best of our knowledge, no multistage network has yet been published that combines the three transformations rigid, affine and deformable. Therefore, our proposed multistage network represents a new approach to medical image registration and addresses the challenge of larger rigid deformations beyond the typical capture ranges of affine registration approaches [29,34] that might occur in medical images.

A fair comparison of the results of our proposed rigid-affine-deformable multistage network with results in papers from published multistage approaches is difficult to achieve. In the published papers, different datasets, image modalities and target structures were used than in our experiments. The differences in the performance of the published multistage methods and our proposed multistage network could therefore also result from the different datasets and modalities used. Moreover, different target structures such as femur [14], brain [13,15,17,21,24–28], liver or kidney (used in our experiments) contain different degrees of rigid, affine or deformable transformations, which may affect the performance of the multistage methods. A fair comparison is therefore only possible if all multistage approaches are evaluated on the same datasets, e.g., by also evaluating the published multistage methods on the datasets used in our experiments. The performance of the published networks on our datasets then also depends on the generalizability of the networks to new data. A comparative study on affine medical image registration networks [37] has shown that only a few networks are generalizable to new data and applications.

### 5.2. Individual Networks

Our results demonstrate that the rigid subnetwork could correct rigid transformations that could not be corrected by the affine subnetwork alone. Using a rigid-affine network yielded better registration results than using an affine subnetwork without a preceeding rigid subnetwork (Table 4).

The results for Dice coefficient and level of image folding were mixed when comparing our approach to a benchmark network. For the first NIDDK dataset, the Dice coefficient of the benchmark and the proposed network were similar. However, upon further visual evaluation, the proposed multistage network showed a higher overlap of all corresponding structures. For the second NIDDK dataset, the spatial alignment between the benchmark and the proposed network was similar, indicating comparable performance. Both models also achieved a comparable performance (similar Dice coefficients) on the external kidney dataset. However, for the augmented external kidney dataset and the $M^2$OLIE liver dataset, the proposed multistage network significantly outperformed the benchmark and yielded significantly ($p < 0.00001$ and $p = 0.00152$) higher Dice coefficients. For the first and second NIDDK dataset, our proposed multistage networks produced registration results with significantly less image folding compared to the benchmark, resulting in medically plausible transformations. For the external kidney dataset, the augmented external kidney dataset and the $M^2$OLIE liver dataset, the proposed network and the benchmark yielded similar levels of image folding. These results show that the proposed multistage network

is promising for improving the registration of certain datasets while competing with the benchmark for other datasets.

The level of image folding is low for all registration approaches used. It should also be noted that the number of Jacobian determinants $\leq 0$ is a global metric. Spatial information, i.e., where image folding occurs, is not represented in this metric. For the neural networks and the baseline, the majority of image folding occurred in areas that are not relevant for the clinical analysis, e.g., in the image background.

The visual evaluation of the deformation fields demonstrated that the deformation within the individual organs and structures was smooth. The deformation fields of the baseline contained large areas in gray. Only in the area of the segmentations or in their surroundings colors are they visible in the deformation field. This is due to the fact that the segmentation of the fixed and moving image was used to determine the structure of interest for the baseline. Outside of these structures, deformations were generated that were identical in all three spatial directions.

*5.3. Individual Datasets*

In the first NIDDK dataset, the T2-weighted MR scans had a slice thickness of 9 mm, three times that of the T1-weighted MR scans. The T2-weighted slice in Figure 3 is therefore interpolated from multiple slices. The multistage network therefore not only performs registration of the T2-weighted scans on the corresponding T1-weighted scans, but also reconstructs thinner (3 mm) T2-weighted slices using T1-weighted MR scans.

We applied the networks trained on the first NIDDK dataset to the second NIDDK dataset, without finetuning the networks with the second dataset. This yielded registration results with a high spatial alignment, which indicates that generalization to new data was successful.

In the external kidney dataset, kidney segmentations were created for the T1-weighted scans, whereas kidney cyst segmentations were created for the T2-weighted MR scans. A Dice coefficient of 100% cannot be achieved even with perfect registration and alignment of the kidneys as the two types of segmentation focus on different aspects. Kidney segmentations cover the entire kidney, including cysts and the renal parenchyma, whereas kidney cyst segmentations exclusively mark the cysts themselves. Moreover, a significant increase in the Dice coefficient does not necessarily indicate better spatial alignment. Hence, a visual evaluation of the registration results is important.

We also applied the networks trained on the first NIDDK dataset to the external kidney dataset. Good registration results were achieved without needing to finetune the networks. Compared to this, the multistage network yielded registration results with a higher Dice coefficient on the augmented external kidney dataset (last row in Table 6). This is due to the fact that the multistage network trained on the first NIDDK dataset was applied to the external kidney dataset without finetuning, whereas the multistage network was finetuned on the augmented external kidney dataset before evaluation on the augmented external kidney dataset. Without finetuning, the multistage network performed poorly on the augmented dataset. The reason for this is that the augmentation generated transformations, such as rotation or translation, which do not occur to this extent in the training dataset (first NIDDK dataset). Hence, finetuning of the multistage network with the augmented external kidney dataset was necessary in order to be able to correct these deformations as well.

We augmented the external kidney dataset, i.e., performed artificial transformations similar to those used to augment training datasets, i.e., rotation, translation, scaling and deformation. In further work, the deformation of the abdomen due to respiration could also be simulated in order to investigate to what extent this deformation can be corrected with the proposed multistage network.

For jointly finetuning all subnetworks of the benchmark and multistage network with the augmented external kidney dataset, we used a large value for the lambda parameter (10) to regularize the deformation field. If the networks were trained with smaller values for the lambda parameter, the networks produced deformations of the moving image that

were not medically plausible. Using a large value for the lambda parameter then resulted in a very low level of image folding, which leads to a medically plausible transformation.

Similarly, for the M$^2$OLIE liver dataset, a large lambda parameter (4.5) was chosen to regularize the deformation field. However, medically implausible deformations were visible in the moved image at one area. These can be prevented by increasing the lambda parameter when finetuning the benchmark and proposed multistage network. However, this then leads to a worse registration result, i.e., lower overlap of all corresponding structures, compared to finetuning with a lambda parameter of 4.5. For the finetuning of the networks, a compromise had to be chosen between a smooth deformation field, which leads to medically plausible transformations (large lambda parameter) and a high deformation correction, i.e., improved spatial alignment through registration (lower lambda parameter, e.g., 4.5).

The segmentations of the kidney datasets were created manually. In contrast, the liver segmentations of the M$^2$OLIE liver dataset were generated by neural networks. The performance of the segmentation networks and the differences in the performance of the CT and MR segmentations could have a significant impact on segmentation-based metrics, and thus on the evaluation with the Dice coefficient.

For the registration of new datasets, the proposed multistage network trained on the first NIDDK dataset can be applied. Finetuning the network with the new dataset can improve the registration result. For this, different stages of the proposed multistage network can be finetuned separately (see finetuning with the augmented external kidney dataset in Section 4.2) or all stages can be finetuned jointly (see finetuning with the M$^2$OLIE liver dataset, Section 4.3).

*5.4. Limitations*

Evaluating image registration is a challenging task, mainly since ground truth data are usually not available [6]. We applied a combination of different approaches for evaluation. If segmentations of organs or structures are available in the fixed and moving image, segmentation-based metrics such as the Dice coefficient can be used to quantify the overlap between the segmentations in the two images before and after registration. Structural differences can be determined visually by color overlay of the fixed and moved images and segmentations (if available). In addition, the deformation field was examined. By calculating the Jacobian determinant of each point in the deformation field, the image folding can be determined. And optical evaluation of the deformation field visualizes the regions with smooth or large deformations.

Training the multistage network with three subnetworks requires a lot of GPU memory. Training the multistage network was only possible with a maximum image size of $256 \times 256 \times 64$ (resolution $2 \times 2 \times 4$ mm$^3$) on an NVIDIA RTX A6000 GPU with 48 GB of GPU memory. This could limit the registration result, especially in the deformable network, as the higher the resolution of the images, the smaller the deformations that can be corrected. In future work, a multistage network could be developed that combines rigid, affine and deformable transformation within a single network without multiple subnetworks (for example, a U-Net-like network), which could be trained with a larger image size (e.g., $512 \times 512 \times 128$, resolution $1 \times 1 \times 2$ mm$^3$) due to the smaller network size.

## 6. Conclusions

We proposed a multistage neural network for three-dimensional multimodal medical image registration which combines rigid, affine and deformable transformations. Our results show that incorporating rigid, affine and deformable transformations within a multistage network yields registration results with a high structural similarity, an overlap of all corresponding structures and a low level of image folding, resulting in a medically plausible transformation.

**Author Contributions:** Conceptualization, A.S. and F.G.Z.; methodology, A.S.; software, A.S.; validation, A.S.; formal analysis, A.S.; investigation, A.S.; resources, A.C. and F.G.Z.; data curation, A.S. and A.C.; writing—original draft preparation, A.S. and F.G.Z.; writing—review and editing, A.S., A.C. and F.G.Z.; visualization, A.S.; supervision, F.G.Z.; project administration, F.G.Z.; funding acquisition, F.G.Z. All authors have read and agreed to the published version of the manuscript.

**Funding:** This research project is part of the Research Campus M$^2$OLIE and funded by the German Federal Ministry of Education and Research (BMBF) within the Framework "Forschungscampus: public-private partnership for Innovations" under the funding code 13GW0388A. This project was supported by the German Federal Ministry of Education and Research (BMBF), under the funding code 01KU2102, and the Italian Ministry of Health, under the frame of ERA PerMed (ERAPERMED2020-326-RESPECT). MRI data collection (dataset 3) was funded in part by ERA—EDTA (Euro-CYST Initiative). For the publication fee we acknowledge financial support by Deutsche Forschungsgemeinschaft within the funding programme "Open Access Publikationskosten" as well as by Heidelberg University.

**Institutional Review Board Statement:** The study was conducted according to the guidelines of the Declaration of Helsinki and approved by the Institutional Review Board "Ethics Committee II" of the Medical Faculty Mannheim, Heidelberg University (2016-863R-MA, 17 November 2016).

**Informed Consent Statement:** Patient consent was waived due to the retrospective nature of data aggregation with no connection to clinical patient care.

**Data Availability Statement:** Our source code is available at https://github.com/Computer-Assisted-Clinical-Medicine/Multistage_Network_for_Rigid_Affine_Deformable_Medical_Image_Registration (accessed on 24 July 2023). The first and second NIDDK dataset were obtained from NIDDK at https://repository.niddk.nih.gov/studies/crisp1/ (accessed on 10 December 2021).

**Acknowledgments:** The Consortium for Radiologic Imaging Studies of Polycystic Kidney Disease (CRISP) was conducted by the CRISP Investigators and supported by the National Institute of Diabetes and Digestive and Kidney Diseases (NIDDK). The data and samples from the CRISP study reported here were supplied by the NIDDK Central Repositories. This manuscript was not prepared in collaboration with investigators of the CRISP study and does not necessarily reflect the opinions or views of the CRISP study, the NIDDK Central Repositories or the NIDDK. We are thankful to the NIDDK for providing us with the patient data from the CRISP study.

**Conflicts of Interest:** The authors declare no conflict of interest.

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
