# Peer review of "A Multistage Rigid-Affine-Deformable Network for Three-Dimensional Multimodal Medical Image Registration"

_applsci, doi:10.3390/app132413298_

Round 1

Reviewer 1 Report

Comments and Suggestions for Authors

Paper summary

* The paper proposes a novel three-stage neural network for multimodal medical image registration. The network combines rigid, affine, and deformable transformations in three stages to address the challenge of larger rigid deformations commonly present between medical images.

* The network is trained in an unsupervised manner using mutual information and gradient L2 loss. It is evaluated on four 3D multimodal datasets - three renal MRI datasets (T1- and T2-weighted) and one liver CT and MRI dataset.

* Results show the network achieves good structural similarity and overlap between registered images, with low folding artifacts. The rigid stage helps capture larger displacements compared to a network with just affine and deformable stages. Experiments demonstrate combining rigid, affine and deformable transformations in a multistage network leads to accurate and medically plausible registrations.

* The proposed network could aid clinical applications like diagnosis and treatment planning by enabling improved fusion of complementary information from different imaging modalities.

Parts I like about this paper:

It applies a new NN architecture for medical imaging across three stages, which helps the network handle larger rigid image deformations compared to prior networks with just affine and deformable stages.

The evaluation shows the network on four diverse 3D multimodal medical image datasets - renal MRI and liver CT/MRI scans. Demonstrates its effectiveness across modalities and body regions. and the experiments comparing several network variants using appropriate evaluation metrics and baselines.

Parts that I don't like

Lack of comparison to other state-of-the-art deep learning based registration techniques. Comparing performance to recent networks could better showcase advantages.

Comments on the Quality of English Language

Easy to understand

Reviewer 2 Report

Comments and Suggestions for Authors

Dear colleagues,

thank you for your quite a remarkable research.

First of all, I would like to highlight the scientific rigorous and solid ground evidences supporting your findings. Undoubtedly, the results are very important for both clinical and research applications providing effective tools to register medical images with high level of certainty and reproducibility.

The only question I would like to discuss is the sentence:  "In contrast, neural networks offer a faster and more efficient approach to image
registration. Instead of solving an optimization problem for each new pair of images, a neural network takes two n-dimensional images as input and outputs the transformation in a single forward pass through the neural network." 

If this is your  theoretical suggestion or experimental finding, please, include the original data. Otherwise, I would recommend providing references of the same quality as your results are.

Yours sincerely,

Reviewer 3 Report

Comments and Suggestions for Authors

The following aspects should be addressed before any further processing may take place.

1. It is not clear whether the proposed approach is able to scale up to process real-world use case settings with large amounts of data. The authors should comment on this, and sustain their assertions with experimental data or, at least, consistent conceptual remarks. 

2. It is not clear how the proposed approach compares against similar existing approaches. Therefore, the authors should add a separate sub-section, which should comparatively analyze the proposed model against 4-5 of the most similar existing approaches. This should highlight the advantages and drawbacks of the proposed approach. The assertions should be sustained by numerical/experimental data and clear conceptual remarks.

3. The mathematical expressions should be linked in a clearer fashion to their algorithmic, functional role, so that the paper's presentation becomes clearer even for readers that are not necessarily proficient in mathematics.

4. The English language should be improved through, at least, one round of proofreading.

Comments on the Quality of English Language

The English language should be improved through, at least, one round of proofreading.

Reviewer 4 Report

Comments and Suggestions for Authors

The presentation stile should be improved. 

1. Please, correct the misprints like on line 131: 2a)),

2. line 98: "by a deformation (displacement) field u"

line 167:  "the deformation field Ï•:" 

In Expressin 5 the variables u and p are not determined.

3. line 208: "mean inter-user agreement of 0.92 ± 0.02 (Dice) with a coefficient of variation of 0.03" this is redundant information since the third value is calculated from the first two. 

4. Section 3.3 should provide more experimental results (even visual examples).

5. The sizes of the images are very small to be able to evalute the accuracy. The ROI should be added.

6. line 352: " ...multistage network: 4.8 ± 2.1, benchmark: 6.1 ± 2.1)" it is not possile to understand the values meaning.

7. From the description of Exp. I (line 271): "First, we trained our proposed multistage rigid-affine-deformable network with the 271 first NIDDK dataset ....(line 279) We then applied the trained networks on the second NIDDK dataset."

In the Results section, the caption to Figure 3 says 

"Fig 3. Example results of the registration of two volumes from the first NIDDK dataset."

Could you explane in more details what do you compare in Fig?

Round 2

Reviewer 3 Report

Comments and Suggestions for Authors

I appreciate that the enhancements applied warrant further consideration of this paper.

Comments on the Quality of English Language

English language is generally fine.

Author Response

Thank you very much for the feedback.

We improved the English language by having the manuscript proofread by two colleagues who are fluent in English.